# Effects of Thermoforming on the Mechanical, Optical, Chemical, and Morphological Properties of PET-G: In Vitro Study

**DOI:** 10.3390/polym16020203

**Published:** 2024-01-10

**Authors:** Edoardo Staderini, Giuseppe Chiusolo, Federica Guglielmi, Massimiliano Papi, Giordano Perini, Michele Tepedino, Patrizia Gallenzi

**Affiliations:** 1School of Dentistry, Dean: Prof. Massimo Cordaro, Catholic University of the Sacred Heart, IRCCS “A. Gemelli” University Polyclinic Foundation, Largo Agostino Gemelli 8, 00168 Rome, Italy; edoardo.staderini@unicatt.it (E.S.); giuseppechiusolo@libero.it (G.C.); patrizia.gallenzi@unicatt.it (P.G.); 2Postgraduate School of Orthodontics–Director: Prof. Massimo Cordaro, Catholic University of the Sacred Heart, Largo Agostino Gemelli 8, 00168 Rome, Italy; 3Department of Neurosciences, Catholic University of the Sacred Heart, Largo Francesco Vito 1, 00168 Rome, Italy; massimiliano.papi@unicatt.it (M.P.); giordano.perini@unicatt.it (G.P.); 4IRCCS “A. Gemelli” University Polyclinic Foundation, Largo Agostino Gemelli 8, 00168 Rome, Italy; 5Department of Biotechnological and Applied Clinical Sciences, University of L’Aquila, Piazza Santa Margherita 2, 67100 L’Aquila, Italy

**Keywords:** thermoforming process, clear aligners, thermoplastic materials, glycol-modified polyethylene terephthalate, surface roughness

## Abstract

The effectiveness of clear aligners in correcting malocclusions is closely linked to the properties of the materials used to make them. The polymers used in the manufacture of clear aligners have well-established properties. However, the process of manufacturing clear aligners, known as thermoforming, involves thermal and mechanical shocks that may alter these properties. The objective of this study was to evaluate the effects of thermoforming on the mechanical, optical, chemical, and morphological properties of sixty PET-G specimens. The study compared the thickness, weight, absorbance, chemical structure, surface roughness, elastic modulus, yield strength, and breaking load of thirty thermoformed specimens with thirty non-thermoformed specimens. The study introduces a new approach by using standardized samples to analyze both chemical and physical properties. The results showed statistically significant differences in thickness (−15%), weight (−11%), and surface roughness (+1233% in roughness average; +1129% in RMS roughness) of the material. Additionally, a correlation was found between reduction in thickness and increase in opalescence. There was no significant change in the functionality of the aligners after thermoforming, as no significant mechanical changes were found. However, the increase in surface roughness may lead to plaque and fluid accumulation and worsen the fit of the aligners.

## 1. Introduction

Clear aligners are orthodontic devices made of thermoplastic material; based on clinical planning, they exert forces on the dental elements in the three planes of space [1,2]. The ability of clear aligners to move dental elements is closely related to the maintenance of the mechanical properties of the material from which they are made [3,4]. Several studies have shown that these mechanical properties are altered both by prolonged exposure to the intraoral environment and thermoforming [5,6,7]. Thermoforming is the process of heating a thermoplastic sheet to its softening point; this allows a clear aligner to be moulded by conforming the thermoplastic sheet to the crown surface of dental models [8,9]. 

In terms of morphological properties, Ryokawa et al. have shown how different materials used in the manufacture of orthodontic aligners undergo variations in thickness after the thermoforming process; their study showed that the least stable materials were copolyesters (Essix A+) and polycarbonates (PC), with a reduction in thickness of 25.1% and 19.8%, respectively, while the most stable were polypropylene (PP) and glycol-modified polyethylene terephthalate (PET-G), with a reduction in thickness of 7.4% and 12.3%, respectively [10]. 

Regarding the optical and mechanical properties, Ryu et al. compared various materials with different thicknesses by evaluating the changes after thermoforming: PET-G specimens with a thickness of 1 mm suffered an about 2% reduction in transparency, an increase in water absorption of approximately 4 µg/mm^3^, and a decrease in the elastic modulus of about 160 Mpa [11]. Golkhani et al. also showed that the thermoforming process leads to a reduction in the elastic modulus, regardless of the type of material, as well as a reduction in the thickness of the specimens (0.1 mm reduction for PET-G) [12]. In contrast, Tamburrino et al. showed an 11% increase in the elastic modulus and a 9% increase in the yield strength of PET-G after thermoforming [13].

The heterogeneity of the results could be attributed to the influence of sample geometry on mechanical properties: Ryu et al. used trapezoidal prism-shaped samples, Golkhani et al. used rectangular sheets, Tamburrino et al. used “dumbbell-shaped” samples of polyethylene terephthalate (PET), glycol-modified polyethylene terephthalate (PET-G), and thermoplastic polyurethane (TPU); TPU has been shown to have higher weight variation but better elastic modulus stability than PET-G. Indeed, Elkholy et al. analysed the changes in the mechanical properties of PET-G samples with different geometries (stone model base plate, stone round disc, gable roof shaped specimen, stainless steel model holding plate) after thermoforming; they showed how the geometry of the material plays a crucial role in its ability to maintain mechanical properties [14]. Therefore, it is important to investigate the stability of mechanical properties using standardised specimens [15].

A deep understanding of thermoforming processes can assist clinicians in selecting the most suitable material and protocol for clear aligner therapy, without exerting excessive pressure on the periodontal ligament [16,17]. In addition, a thorough comprehension of the material’s behaviour can enhance the accuracy levels of clear aligners, which currently range from 50 per cent to 73.6 per cent [18,19].

The aim of the present in vitro study is to analyse the impact of the thermoforming process on the mechanical, optical, chemical, and morphological properties of standardised PET-G “dumbbell-shaped” samples. 

The novelty of this study concerns the integration of both chemical and mechanical testing procedures, while most studies in the literature analysed only either chemical or physical properties [15,16,17]. Additionally, all these assessments were conducted on ISO samples. Furthermore, the study concentrated on detecting the morphological variations of the material concerning surface roughness, which has been scarcely investigated in the literature thus far. In fact, the thermal shock associated with the thermoforming process could alter the surface roughness and thus partially affect the optical and absorption properties of the material [20,21]. Lira et al. found an increase in surface roughness after 14 days of intraoral use, resulting in a loss of transparency and increased pigmentation of the aligners [22]. However, no study has yet investigated the effects of thermoforming on surface roughness [21,23].

## 2. Materials and Methods

At baseline, 60 PET-G sheets (Erkodur, Erkodent Erich Kopp GmbH, Pfalzgrafenweiler, Germany) were selected and randomly assigned by a third person (not involved in the research) to two different groups: 30 sheets (thermoforming group—TG) underwent the thermoforming process, while 30 sheets (control group—CG) were analysed as supplied.

The thermoforming process was carried out using a Ministar S machine (Ministar S, Scheu, Iserlohn, Germany) according to the manufacturer’s recommendations for pressure, heating time, and cooling time. Specifically, a pressure of 4/4.2 bar, a temperature of 175 ± 10°, a heating time of 28″, and a cooling time of 30″ were applied. The disk temperature was measured at the surface using a laser thermometer (Fluke 62 Max +, Fluke, Everett, Washington, DC, USA). Every PET-G sheet was vacuum thermoformed without the use of any mould to simulate the stress of the thermoforming procedure. From each PET-G sheet, “dumbbell-shaped” specimens were cut using a laser cutting machine (BCL1309X, Bodor, Jinan, China) (Figure 1).

The shape, size, and thickness of the “dumbbell-shaped” specimens followed the EN ISO 527-2 guidelines for type 5B specimens. To ensure consistency in mechanical and optical measurements, stringent experimental protocols were adhered to throughout the testing process.

The morphology of the specimens consists of two shoulders (at the ends) and a gauge section (in the middle). The shoulders are wider than the gauge section, resulting in a stress concentration in the middle when the sample is subjected to a tensile load. In physics, if a specimen fails in the middle section, it is due to the material reaching its maximum tensile strength; if, on the other hand, the specimen fails at one end or in the socket itself, the failure can be attributed to improper loading or a pre-existing defect in the material. The “dumbbell-shaped” morphology is therefore designed to increase the probability of specimen failure at maximum tensile load.

Thickness analysis, weight analysis, absorbance analysis, Fourier-transform infrared spectroscopy (FTIR), surface roughness analysis, and tensile tests were evaluated (Table 1).

### 2.1. Thickness

The thickness of the specimens was measured with a digital electronic gauge (Fervi SpA, Vignola, Italy), with an accuracy of 0.01 mm. Measurements were taken at random points on the shoulders and in the gauge section, and then the mean was calculated.

### 2.2. Weight

Weight was measured using an Entris scale (Entris, Sartorius, Germany) with a readability of 0.1 mg.

### 2.3. Absorbance

Absorbance was measured in the visible light spectrum (frequency range: 230–700 nm; step: 5 nm) through a Cytation3 spectrophotometer (Cytation3 Imaging reader, Biotek, Santa Clara, CA, USA) (Appendix A). This measurement was performed to assess any changes in the transparency of the material after thermoforming. Each specimen was fixed to a customized thermoplastic holder (Polylactic Acid PLA, FILOALFA^®^, Torino, Italy) (Appendix A), and then inserted into a plate (*take 3*) of the machine. The above-mentioned holder was realized using a 3D printer (Creality Ender-3, Longhua Dist., Shenzhen, China). After the measurement, the data were transferred from the machine to specific software (Gen 5, Microplate Reader and Imager Software, Biotek, Santa Clara, CA, USA) capable of collecting and analysing them. All measurements were performed at a constant temperature of 23 °C.

### 2.4. FTIR Spectroscopy

This FTIR spectroscopy was carried out to investigate any changes in the chemical composition of the PET-G material after the thermoforming process. The measurements were carried out using the Alpha II spectrometer (Alpha II, Bruker, Germany) (Appendix A), which is able to detect the absorbance of the samples and to compare the spectrum of the results obtained with an internal software. This test aimed to evaluate whether the thermoforming process leads to changes in the arrangement and distribution of the chemical bonds of the polymer under analysis. Spectral transmittance data were obtained with a frequency range (MIR) of 4000–600 cm^−1^; 24 consecutive scans were performed for each measurement. All measurements were made on one shoulder of the sample as the chemical composition of the material is expected to be the same at each point in the sample.

### 2.5. Surface Roughness

Surface roughness was assessed using an atomic force microscope (JPK Nanowizard II, Bruker, Germany) (Appendix A). The surface roughness parameters recorded were (a) *Ra* (roughness average), the arithmetic average of the absolute values of the profile heights over the evaluation length, and (b) *Rq* (RMS roughness), the root mean square (RMS) average of the profile heights over the evaluation length. Measurements were randomly taken at multiple sample points and then the mean was calculated.

### 2.6. Tensile Tests

Tensile tests were carried out with a Univert machine (Univert, Cell Scale, Waterloo, ON, Canada) in accordance with EN ISO 527-1:2019 (Appendix A). The Univert machine allowed for evaluation of the tensile strength of the materials by means of two parameters: the displacement of the two clamps per unit of time, or the force between the two clamps per unit of time. In accordance with the literature, the first setting was chosen for this study, as it allowed for evaluation of the force required to achieve a given effect on the material. Two different strain rates were used: the slow dynamic (0.8 mm/s) allowed us to analyse the yield strength; the fast dynamic (8 mm/s) allowed us to analyse the breaking load and the elastic modulus (Young’s modulus). 

### 2.7. Statistical Analysis

All data are expressed as mean ± standard deviation.

Dalaie et al. have shown that a sample size of 5 specimens was required to obtain a power of the study equivalent to 0.8, considering an α error of 0.05 [6]. In this experimental study, both non-destructive and destructive tests were performed. A total of 30 samples were selected for non-destructive testing (thickness, weight, absorbance, FTIR spectroscopy, surface roughness). Instead, the analysis of the mechanical properties consisted of destructive testing; therefore, 15 specimens were employed for the measurement of the yield strength and 15 for the measurement of the breaking load and the elastic modulus. Independent *t*-tests were performed for the synchronous comparison of the material before and after the thermoforming process. The inter-operator reproducibility of the results has been systematically verified by conducting *t*-tests on a cohort of 30 samples. All the procedures were meticulously controlled and repeated by one researcher (M.P.) to validate the intra-operator reliability and replicability of the obtained outcomes. The significance level was set at *p* < 0.01. All data were analysed using Systat software (version 8.0, SYSTAT Software Inc. (SSI), San Jose, CA, USA). 

## 3. Results

### 3.1. Reliability

The paired *t*-test confirmed the intra- and inter-operator reproducibility of the observed effects across the sample set.

### 3.2. Thickness

As shown in Table 2, the thermoforming process resulted in a 15% reduction in the thickness of the specimens (*p* < 0.01); the height of the specimens remained unchanged.

### 3.3. Weight

As shown in Figure 2 and Table 3, the average weight of the samples was reduced by approximately 11% after thermoforming (*p* < 0.01). 

### 3.4. Absorbance

As shown in Figure 3, no significant differences in the optical density of the TG samples could be detected as the absorbance spectra overlapped. According to the Beer–Lambert law, light attenuation depends on the properties of the material through which the light is passing:A=εl
where:

A is the absorbance.

ε is the molar attenuation coefficient or absorptivity of the attenuating species. 

*ℓ* is the optical path length in cm (thickness).

Since the absorbance values were similar between CG and TG specimens, but the thickness of the TG specimens was reduced after thermoforming, we should expect that there was an inversely proportional increase in the molar attenuation coefficient.

### 3.5. FTIR Spectroscopy

As shown in Figure 4, there were no significant chemical changes in the samples after treatment.

### 3.6. Surface Roughness

As shown in Figure 5a,b, the material underwent important alterations in surface roughness after thermoforming.

As shown in Figure 5c,d and in Table 4, the thermoforming process resulted in an increase of approximately 1233% in the roughness average, which is the arithmetic average of the absolute values of the profile heights over the evaluation length, and an increase of approximately 1129% in the RMS roughness, which is the root mean square average of the profile heights over the evaluation length.

### 3.7. Tensile Tests

As shown in Table 5, no statistically significant differences in mechanical properties between CG and TG samples were appreciated (*p* > 0.01). TG samples were found to have inferior mechanical properties solely due to the variation in thickness. 

Both groups (CG and TG) showed the same mechanical behaviours under destructive tensile tests. The fast dynamics showed that the breakage of the molecular bonds of the polymer sample occurred after about 6% of elongation (Figure 6a). With the slow dynamics, it was not possible to observe the breakage of the molecular bonds of the polymer but only a slow elongation of its fibres (Figure 6b). 

## 4. Discussion

### 4.1. Interpretation of the Results in the Context of the Available Literature

In this experimental study, the mechanical, chemical, optical, and morphological properties of PET-G, one of the most used materials for clear aligners, were analysed. Several studies in the literature have attempted to evaluate whether this process could lead to significant alterations in the material’s properties, as it is known that the maintenance of its properties is fundamental to keep its ability to move teeth [24,25].

Dalaie et al. showed that thermoforming significantly (*p* < 0.01) reduced the flexural modulus, hardness, elastic modulus, and glass transition temperature of PET-G sheets (Erkodur, Erkodent Erich Kopp GmbH, Pfalzgrafenweiler, Germany) [6]. Golkhani et al. also analysed the variations in PET-G sheets following the thermoforming process and found a statistically significant reduction in both elastic modulus (557 MPa, from 2746 to 2189 MPa) and thickness (0.10 mm) [12]. Similarly, Ryu et al. also observed a significant reduction in thickness of PET-G trapezoidal prism specimens (*p* < 0.01) [11].

Although the present study involved the use of standardised “dumbbell-shaped” specimens, our results are consistent with those of Golkhani et al. and Ryu et al., as the specimens underwent a significant average thickness reduction of 0.15 mm (*p* < 0.01) after thermoforming. These results agree with a study by Palone et al., who used 2D methods to assess the thickness variation of different types of aligners after thermoforming: they showed that regardless of the material, thermoforming significantly reduces thickness of the aligners [16,26].

This reduction in thickness could be the main cause of the subtle changes in the mechanical properties of the material [11,12,27]. Indeed, there was a slight reduction in the elastic modulus of the material, although this reduction was not significant (*p* > 0.01), while the studies by Dalaie et al. and Golkhani et al. on PET-G sheets showed a significant reduction [6,12]. 

In contrast to the present data, Tamburrino et al. showed a significant increase in the elastic modulus of dumbbell-shaped specimens after thermoforming (*p* < 0.05), and they attributed this increase to a process known as “drawing”, which occurs when the material is heated and stretched: the polymer chains slide over each other, thus orienting part of the chains along the direction of the force [13].

As a possible explanation for the conflicting data on the variation of elastic modulus after thermoforming, FTIR analysis showed that the thermoforming process did not change the polymeric structure of the specimens. In physics, the elastic modulus of a sample would be expected to change if any variation occurred in the crystalline/amorphous ratio of the polymer structure. In this case, there were no statistically significant differences in the elastic modulus of the TG and CG group, as expected.

When analysing stress–strain curves, it is important to note the impact of pulling speed on the polymer’s mechanical behaviour. Our data shows that both groups (CG and TG) exhibit similar mechanical behaviours during destructive tensile tests. However, fast dynamics reveal that the polymer sample’s molecular bonds break after approximately 6% elongation. In contrast, slow dynamics only allow for the observation of a slow elongation of the polymer fibres, without the possibility of observing the rupture of molecular bonds. These findings emphasize the critical role of the rate of force application in determining the mechanical properties of the polymer [28,29]. Rapid breakage indicates increased material brittleness, suggesting that the rupture process occurs abruptly when subjected to high stresses. On the other hand, slow pulling results in a more gradual deformation, highlighting the influence of pulling speed on the polymer’s mechanical response under tensile stress. This nuanced understanding enhances the characterization of the polymer’s mechanical properties, illuminating its behaviour under various loading conditions and offering valuable insights for practical applications and material design considerations.

Regarding the optical properties, this study showed that the reduction in thickness as well as the increase in surface roughness may have indirectly played a role in increasing the optical density of the samples. 

In fact, the analysis of the surface roughness of the specimens showed very interesting data: the average roughness and the RMS roughness increased by about 1233% and 1129%, respectively, after thermoforming (Figure 5c). This increase in surface roughness could explain the discrepancy between mass and thickness reduction (10% and 15%, respectively) and could also be at the basis of the differences in absorbance between CG and TG specimens. 

In our opinion, increased surface roughness, combined with a lack of proper oral hygiene, may have detrimental effects on plaque retention and the absorption of water, saliva, and other fluids, leading to discoloration of clear aligners during intraoral use [21,22]. 

However, further investigation is required to determine the clinical consequences of surface roughness alterations in the physiological process of fluid and debris absorption, as well as plaque accumulation [30,31].

As for the chemical properties, however, these do not appear to be altered by the thermoforming process.

### 4.2. Strengths of the Study

This study was carried out on samples that followed the European ISO guidelines for thermoplastic samples. Moreover, all tests were performed on 30 specimens, except for mechanical measurements which were performed on 15 specimens. Previous studies by Ryu et al. and Dalaie et al. performed sample size analysis enrolling 4 and 5 samples, respectively [6,11]. Since our study showed conflicting data on elastic modulus with those provided by Dalaie et al., it was chosen to increase the sample size by approximately 150% to enhance the study’s statistical power [6].

### 4.3. Limitations of the Study

A limitation of this study could be related to the fact that all the tests were carried out exclusively on “dumbbell-shaped” samples and not on the aligners themselves; however, the shape of the aligners is not homogeneous as they adapt to the shape of the patient’s arch form, thus hampering the reproducibility of the data.

There are only a few studies in the literature that used aligners as samples for the evaluation of optical properties [7,32]. An in-house pilot analysis (data not shown) revealed that using aligners as specimens could lead to unpredictable results, especially for mechanical and optical tests.

## 5. Conclusions

The present in vitro study aimed to assess the mechanical, optical, chemical, and morphological properties of PET-G dumbbell-shaped specimens after the thermoforming process. The novelty of this study resides on the comprehensive analysis of both chemical and mechanical testing procedures, combined with the use of ISO samples, and the analysis of surface roughness. The research methodology investigated the surface roughness, as it plays a critical role both for optical and absorption properties of aligner materials. 

The findings of the present study can be summarized as follows:(a)Morphological changes: the thickness of the TG specimens was reduced by 15% and the molecular weight by 11%; the surface roughness showed a statistically significant increase.(b)Optical changes: the material seemed to be more opaque, probably due to the surface roughness increase that occurs after thermoforming.(c)There were no chemical changes in the material.(d)The mechanical properties of the PET-G dumbbell-shaped specimens remain almost unchanged after thermoforming; the slight differences in mechanical behaviour seem to be related to the thickness reduction of the specimens after thermoforming.

These findings suggest that the mechanical properties of PET-G material, and consequently the clinical performance of the aligners, appear to remain stable after thermoforming. However, the thermoforming process altered the morphological (molecular weight, thickness, and surface roughness) and optical properties (light absorbance) of PET-G material, thus affecting the aesthetics of clear aligners. Future studies should investigate the clinical consequences of the increase in surface roughness. It may also be important to assess whether overcorrection of the initial material thickness is necessary to compensate for the reduction that occurs after thermoforming.

## Figures and Tables

**Figure 1 polymers-16-00203-f001:**
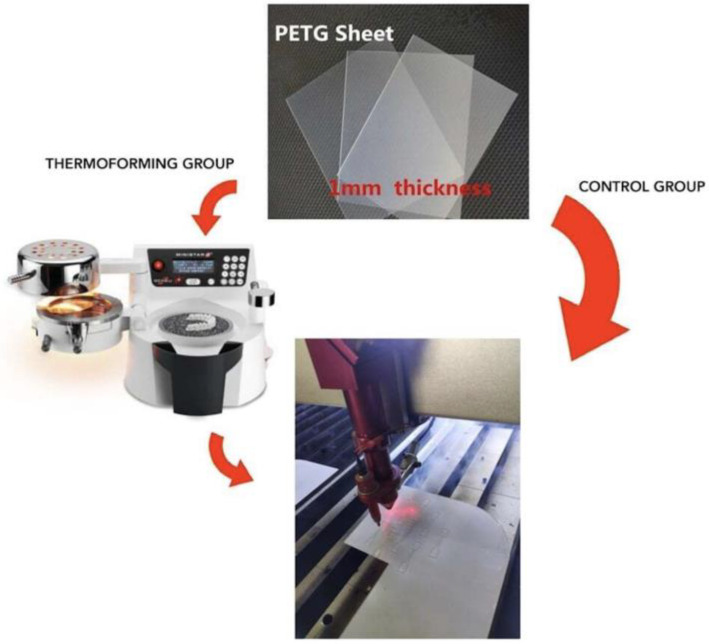
CG and TG sample making process.

**Figure 2 polymers-16-00203-f002:**
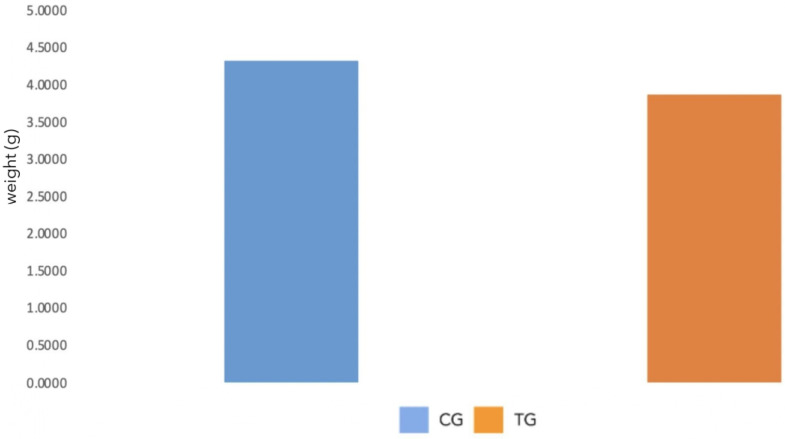
Average weight of the CG (blue) and TG (orange) samples.

**Figure 3 polymers-16-00203-f003:**
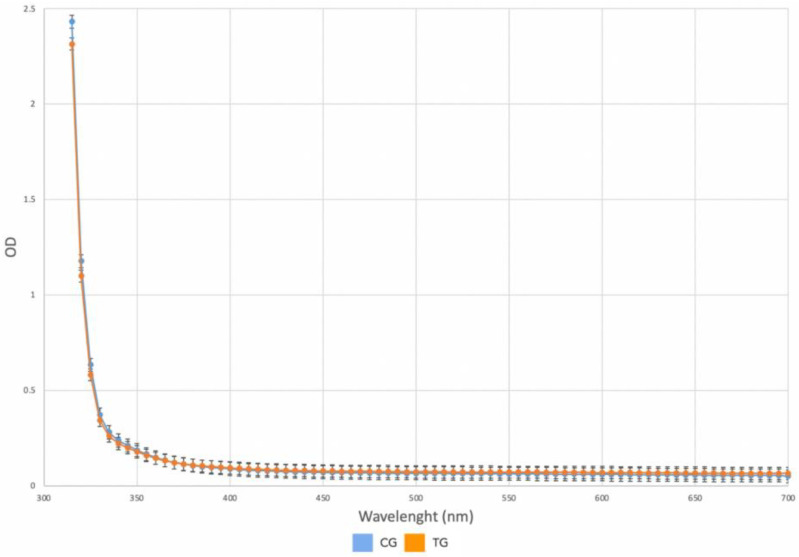
Absorbance spectra of the CG and TG samples; average absorbance CG (blue); average absorbance TG (orange).

**Figure 4 polymers-16-00203-f004:**
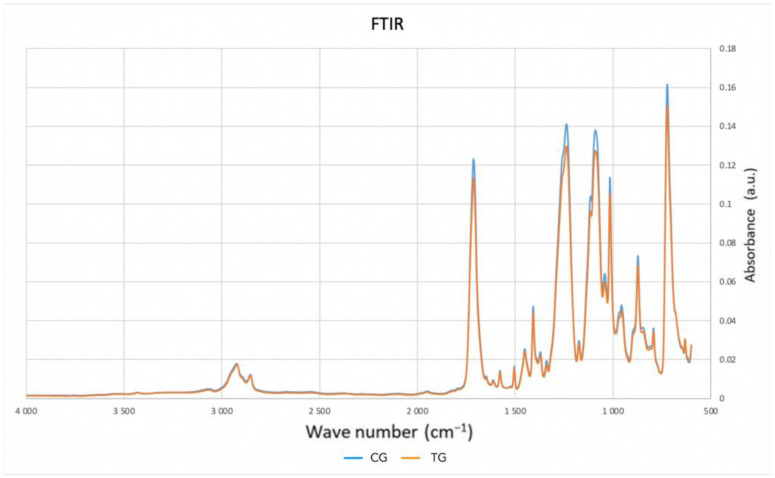
Average FTIR spectroscopy spectrum of the CG (blue) and TG (orange) samples.

**Figure 5 polymers-16-00203-f005:**
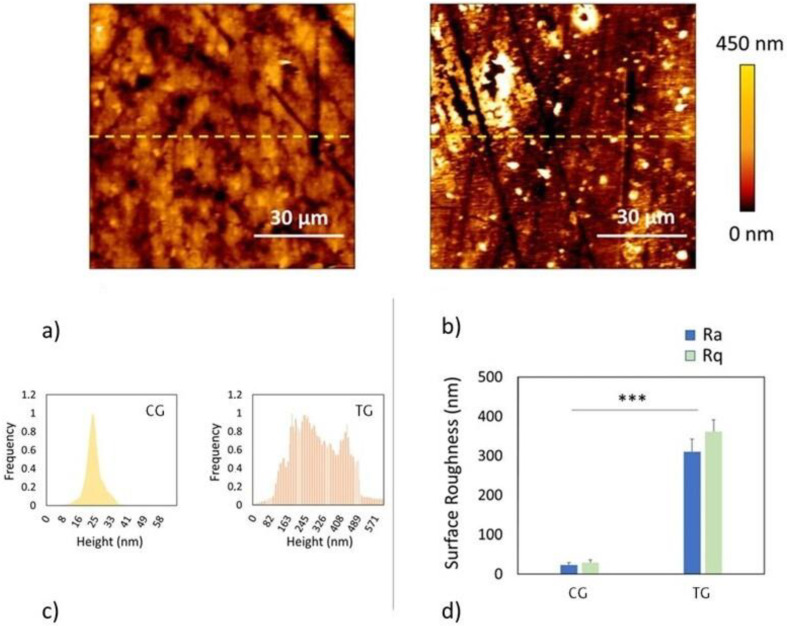
(**a**) Representative surface of the CG sample made with the atomic force microscope; (**b**) representative surface of the TG sample made with the atomic force microscope; (**c**) histogram chart of the profile heights on the surface of a CG sample and a TG sample. “Frequency” refers to how often a given value was observed. The curve is normalized to 1, meaning that the most frequent value is assigned a value of 1 and all other values are proportional; (**d**) quantification of CG and TG roughness based on roughness average (*Ra*) and RMS roughness (*Rq*). * *p* < 0.01; ** *p* < 0.001; *** *p* < 0.0001.

**Figure 6 polymers-16-00203-f006:**
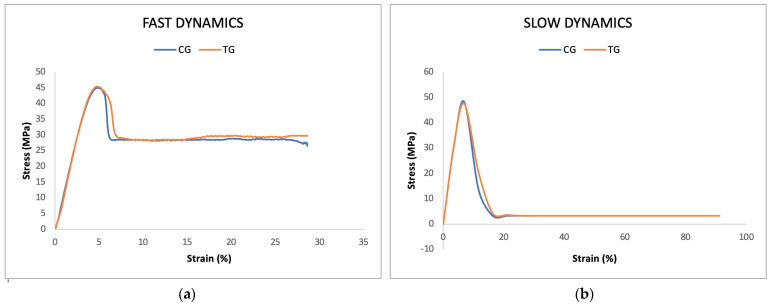
(**a**) Fast dynamics: representative stress–strain curve (8 mm/s) of the CG (blue) and TG (orange) samples; (**b**) slow dynamics: representative stress–strain curve (0.8 mm/s) of the CG (blue) and TG (orange) samples.

**Table 1 polymers-16-00203-t001:** Properties analysed and tests performed.

Properties	Test
Morphological properties	Thickness; weight; surface roughness
Optical properties	Absorbance
Chemical properties	Fourier-transform infrared spectroscopy (FTIR)
Mechanical properties	Tensile tests

**Table 2 polymers-16-00203-t002:** Mean values and standard deviations of thickness and height of control group (CG) and thermoforming group samples (TG).

	CG	TG
Thickness (mm)	0.99 (±0.03)	0.84 (±0.01)
Height (mm)	1.84 (±0.02)	1.84 (±0.01)

**Table 3 polymers-16-00203-t003:** Mean values and standard deviations of weight of CG and TG samples.

	CG	TG
Average weight (g)	4.313 (±0.001)	3.859 (±0.001)

**Table 4 polymers-16-00203-t004:** Mean values and standard deviations of roughness average (*Ra*) and RMS roughness (*Rq*) of CG and TG samples.

	CG	TG
Roughness average (*Ra*)	23.28 (±6.145)	310.34 (±31.626)
RMS roughness (*Rq*)	29.38 (±7.067)	361.16 (±30.656)

**Table 5 polymers-16-00203-t005:** Mean values and standard deviations of yield point, Young’s modulus, and breaking load of CG and TG samples.

	CG	TG
Yield point (MPa)	41.3 (±3.8)	42.3 (±1.8)
Young’s modulus (GPa)	0.73 (±0.01)	0.72 (±0.01)
Breaking load (MPa)	47.4 (±1.1)	48.5 (±2.1)

## Data Availability

The datasets used and/or analysed during the current study are available from the corresponding author on reasonable request.

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
