# Peer review of "Effects of Thermoforming on the Mechanical, Optical, Chemical, and Morphological Properties of PET-G: In Vitro Study"

_polymers, 2024, doi:10.3390/polym16020203_

Round 1
Reviewer 1 Report
Comments and Suggestions for Authors
The current manuscript investigates the effect of thermoforming on the mechanical, optical, chemical, and morphological properties of PETG. The direct application of the current study is significant. However, there are some major points to be considered as follows:
- The problem statement should be clearly defined by the end of introduction.
- The material of the 3d printed mould should be reported.
- Figures captions should be carefully revised. All captions should be under the Figure plots.
- The standard deviation of the experimental measurements should be included.
- It is highly recommended to add a Figure displaying the stress strain plot for both CG and CG groups.
- The obtained results and analysis should be justified and validated. There should be a comparison between the results obtained from the current study and those were reported in the literature.
- More recent and significant references should be cited to enrich the introduction and discussion section.
- The conclusion should focus on the main results, contributions, novelty, and future work.
- The writing style should be carefully revised, a moderate language edit is required.
Comments on the Quality of English Language
Moderate change is required.
Author Response
Comments to the Author:
- The current manuscript investigates the effect of thermoforming on the mechanical, optical, chemical, and morphological properties of PETG. The direct application of the current study is significant. However, there are some major points to be considered as follows:
- The problem statement should be clearly defined by the end of introduction.
Response to the Reviewer:
We thank the reviewer for appreciating the paper. As suggested, we modified the problem statement in the revised manuscript (page 2, lines 84-86).
Comments to the Author:
- The material of the 3d printed mould should be reported.
Response to the Reviewer:
We thank the reviewer to take this aspect up. Indeed, we added the name of the 3D printed mould in the M&M section (page 5, line 144)
Comments to the Author:
- Figures captions should be carefully revised. All captions should be under the Figure plots.
Response to the Reviewer:
Yes, it is correct. The figure captions have been updated.
Comments to the Author:
- The standard deviation of the experimental measurements should be included.
Response to the Reviewer:
Point taken.
The reviewer is fully right: the SD have been added in the revised manuscript (tables 2, 3, 4, 5)
We hope we were able to address correctly the comment of the reviewer.
Comments to the Author:
- It is highly recommended to add a Figure displaying the stress strain plot for both CG and CG groups.
Response to the Reviewer:
We would thank the Reviewer for the comment and following his advice, we added a stress-strain plot in the revised manuscript (Figure 6)
Comments to the Author:
- The obtained results and analysis should be justified and validated. There should be a comparison between the results obtained from the current study and those were reported in the literature.
Response to the Reviewer:
Thanks for the kind suggestions.
As suggested, we have updated the generalization of the results in the context of the available literature in the revised manuscript (page 9, lines 278-301).
Comments to the Author:
- More recent and significant references should be cited to enrich the introduction and discussion section.
Response to the Reviewer:
We would thank the Reviewer for the detailed and positive information. We have now updated the references in the revised manuscript. We have cited more up-to-date articles, as the followings:
1 - Eslami, S.; Kopp, S.; Goteni, M.; Dahmer, I.; Sayahpour, B. Alterations in the surface roughness and porosity parameters of directly printed and Invisalign aligners after 1 week of intraoral usage: An in vivo prospective investigation. American journal of orthodontics and dentofacial orthopedics : official publication of the American Association of Orthodontists, its constituent societies, and the American Board of Orthodontics 2023, 10.1016/j.ajodo.2023.07.013, doi:10.1016/j.ajodo.2023.07.013.
2 - Lira, L.F.; Otero Amaral Vargas, E.; Moreira da Silva, E.; Nunes da Silva Meirelles Doria Maia, J.; Elzubair, A.; Siqueira de Morais, L.; Alvaro de Souza Camargo, S., Jr.; Serra, G.; Gomes de Souza, M.M. Effect of oral exposure on chemical, physical, mechanical, and morphologic properties of clear orthodontic aligners. American journal of orthodontics and dentofacial orthopedics : official publication of the American Association of Orthodontists, its constituent societies, and the American Board of Orthodontics 2023, 164, e51-e63, doi:10.1016/j.ajodo.2023.05.015.
3 - Xiang, B.; Wang, X.; Wu, G.; Xu, Y.; Wang, M.; Yang, Y.; Wang, Q. The force effects of two types of polyethylene terephthalate glyc-olmodified clear aligners immersed in artificial saliva. Scientific reports 2021, 11, 10052, doi:10.1038/s41598-021-89425-8.
Comments to the Author:
- The conclusion should focus on the main results, contributions, novelty, and future work.
Response to the Reviewer:
Correct observation; the conclusion has been modified in the revised manuscript (page 11, lines 357-381)
Comments to the Author:
- The writing style should be carefully revised, a moderate language edit is required.
Response to the Reviewer:
Point taken; the language style has been extensively modified in the revised manuscript( as an example: “as supplied” was used instead of “as received”, page 3, line 105)
Reviewer 2 Report
Comments and Suggestions for Authors
This paper presents the evaluation of the changes in mechanical, optical, chemical, and morphological properties of glycol-modified polyethylene terephthalate (PET-G) specimens after thermoforming. The title of the article is new practical and attractive to the reader. The following should also be considered before acceptance.
The abstract can be made more attractive by using more quantitative data. It is suggested to provide more quantitative results.
The novelty and purpose of the research should be clearly stated in the abstract and introduction. The manuscript needs general writing and grammar editing.
What is the advantage of PETG compared to {MMA, which is the most used in this field?
At first glance, the introduction is too short. Also, the first paragraphs presented are primarily general and general information. At the end of the introduction, a suitable summary of the importance of the present issue should be provided. Also, discontinuity between paragraphs is evident in most of the introduction. It is suggested to rewrite the introduction.
Figure 1(b) should be deleted. It is enough to provide the ASTM standard and the conditions of the tensile test.
The number of reviewed and used references is very small. It is also suggested to use the sources that have been published in the last few years.
Convert force-displacement diagrams to stress-strain. In this section, only the results are presented and there is no discussion that needs to be modified.
Add error bar to results. How has the reproducibility of the results been checked?
The results section is well organized and categorized. However, some parts report the results, which require corrections and deepening the analysis and discussion.
The conclusion needs rewriting. In the conclusion section, a summary of the purpose of the research, innovation, and research method should be presented before presenting the highlights.
Comments on the Quality of English Language**
Author Response
Comments to the Author:
This paper presents the evaluation of the changes in mechanical, optical, chemical, and morphological properties of glycol-modified polyethylene terephthalate (PET-G) specimens after thermoforming. The title of the article is new practical and attractive to the reader. The following should also be considered before acceptance.
Response to the Reviewer:
We thank the reviewer for the attention and importance that he has given to our manuscript; we hope that our contribution can improve the knowledge on aligner materials’ properties in orthodontic workflow.
Comments to the Author:
- The abstract can be made more attractive by using more quantitative data. It is suggested to provide more quantitative results.
Response to the Reviewer:
We thank the reviewer to take this aspect up. Indeed, we added more quantitave data in the abstract (page 1, lines 35-37)
Comments to the Author:
- The novelty and purpose of the research should be clearly stated in the abstract and introduction. The manuscript needs general writing and grammar editing.
Response to the Reviewer:
Yes, it is correct. The aim of the study has been clearly stated in the revised manuscript (page 2, lines 64-86); the novelty of the study has been added to the introduction (pages 2-3, lines 87-97).
Comments to the Author:
- What is the advantage of PETG compared to {MMA, which is the most used in this field?
Response to the Reviewer:
Point taken.
The reviewer is fully right: the PMMA has been extensively used for many orthodontic appliances; the most studied commercially available materials for clear aligners’ manufacturers are PET-G and TPU. Tamburrino et al. used “dumbbell-shaped” samples of glycol-modified polyethylene terephthalate (PET), and Thermoplastic polyurethane (TPU); TPU has shown to have higher weight variation but better elastic modulus stability than PET-G. In the study of Tamburrino et al, TPU has shown to have higher weight variation but better elastic modulus stability than PET-G. Hovewer, our data showed that there were no statistically significant differences between thermoformed (TG) and “as supplied” (CG) specimens, thus confirming the stability of PET-G. This point has been discussed in the revised manuscript. (page 9, lines 299-303)
Comments to the Author:
- At first glance, the introduction is too short. Also, the first paragraphs presented are primarily general and general information. At the end of the introduction, a suitable summary of the importance of the present issue should be provided. Also, discontinuity between paragraphs is evident in most of the introduction. It is suggested to rewrite the introduction..
Response to the Reviewer:
We would thank the Reviewer for the comment and following his advice, we modified the introduction in the revised manuscript (page 2, lines 81-85). We hope we were able to address correctly the comment of the reviewer.
Comments to the Author:
- Figure 1(b) should be deleted. It is enough to provide the ASTM standard and the conditions of the tensile test.
Response to the Reviewer:
Thanks for the kind suggestions.
As suggested, EN ISO 527-2 guidelines for type 5B specimens have been added in the revised manuscript (page 3, lines 117-118).
Comments to the Author:
- The number of reviewed and used references is very small. It is also suggested to use the sources that have been published in the last few years.
Response to the Reviewer:
We would thank the Reviewer for the detailed and positive information. More recent and significant references have been cited to enrich the introduction and discussion section in the revised manuscript. We have cited more up-to-date articles, as the followings:
1 - Eslami, S.; Kopp, S.; Goteni, M.; Dahmer, I.; Sayahpour, B. Alterations in the surface roughness and porosity parameters of directly printed and Invisalign aligners after 1 week of intraoral usage: An in vivo prospective investigation. American journal of orthodontics and dentofacial orthopedics : official publication of the American Association of Orthodontists, its constituent societies, and the American Board of Orthodontics 2023, 10.1016/j.ajodo.2023.07.013, doi:10.1016/j.ajodo.2023.07.013.
2 - Lira, L.F.; Otero Amaral Vargas, E.; Moreira da Silva, E.; Nunes da Silva Meirelles Doria Maia, J.; Elzubair, A.; Siqueira de Morais, L.; Alvaro de Souza Camargo, S., Jr.; Serra, G.; Gomes de Souza, M.M. Effect of oral exposure on chemical, physical, mechanical, and morphologic properties of clear orthodontic aligners. American journal of orthodontics and dentofacial orthopedics : official publication of the American Association of Orthodontists, its constituent societies, and the American Board of Orthodontics 2023, 164, e51-e63, doi:10.1016/j.ajodo.2023.05.015.
3 - Xiang, B.; Wang, X.; Wu, G.; Xu, Y.; Wang, M.; Yang, Y.; Wang, Q. The force effects of two types of polyethylene terephthalate glyc-olmodified clear aligners immersed in artificial saliva. Scientific reports 2021, 11, 10052, doi:10.1038/s41598-021-89425-8.
Comments to the Author:
- Convert force-displacement diagrams to stress-strain. In this section, only the results are presented and there is no discussion that needs to be modified.
Response to the Reviewer:
Correct observation: the stress-strain diagram has been added (Figure 6) and the findings have been reported and commented in the revised manuscript (page 10: lines 310-323)
Comments to the Author:
- Add error bar to results. How has the reproducibility of the results been checked?
Response to the Reviewer:
Point taken; the reproducibility of the results has been reported in M&M (page 5, lines 191-195) and in the results section (page 5, lines 199-200)
Comments to the Author:
- The results section is well organized and categorized. However, some parts report the results, which require corrections and deepening the analysis and discussion
Response to the Reviewer:
Correct observation: the findings have been reported and commented in the revised manuscript (pages 6-9)
Comments to the Author:
- The conclusion needs rewriting. In the conclusion section, a summary of the purpose of the research, innovation, and research method should be presented before presenting the highlights.
Response to the Reviewer:
Point taken; the conclusion has been modified as suggested in the revised manuscript (page 11, lines 357-381)
Round 2
Reviewer 1 Report
Comments and Suggestions for Authors
The revised manuscript is significantly improved. The review comments and recommendations are well addressed. However, there some minor issues to be considered as follows:
- There is no need to keep the sentence at line #100.
- There is no need to redefine "PET-G" at line#103.
- The numbering of subsections should be carefully revised as the subsection 3.4 is not existed.
Comments on the Quality of English LanguageMinor edit is recommended.
Author Response
Comments to the Author:
The revised manuscript is significantly improved. The review comments and recommendations are well addressed.
Response to the Reviewer:
We are grateful for the efforts spent by the Reviewers in assessing our manuscript.
Comments to the Author:
However, there some minor issues to be considered as follows:
- There is no need to keep the sentence at line #100.
Response to the Reviewer:
We thank the reviewer to take this aspect up. Indeed, we removed the sentence in the revised manuscript.
Comments to the Author:
- There is no need to redefine "PET-G" at line#103.
Response to the Reviewer:
Yes, it is correct. The sentence has been updated in the revised manuscript (page 3, line 101).
Comments to the Author:
- The numbering of subsections should be carefully revised as the subsection 3.4 is not existed.
Response to the Reviewer:
Point taken. The numbering of subsections has been checked. We also added subsections in the discussion (page 9, line 272; page 10, lines 339 and 347). We hope we were able to address correctly the comment of the reviewer.